# Methodological Proposal for the Accompaniment and Development of Service+Learning Methodology in Universidad de Las Americas in Chile

**DOI:** 10.3390/ijerph19148626

**Published:** 2022-07-15

**Authors:** Mario Fuentes-Rubio, Antonio Castillo-Paredes

**Affiliations:** Grupo AFySE, Investigación en Actividad Física y Salud Escolar, Escuela de Pedagogía en Educación Física, Facultad de Educación, Universidad de Las Américas, Santiago 8370040, Chile; acastillop85@gmail.com

**Keywords:** social education, educational interventions, teacher training, active methodologies university

## Abstract

Service+Learning (S+L) is an active teaching methodology that enables students to participate in their teaching and learning process, being guided by the teacher. This methodology allows the student to show all the skills, abilities, tools and theoretical-practical knowledge since they are placed at the service of the community partner, which in the case of pedagogies translates into pedagogical practice centers. This transfer must be guided and mediated by the teacher, allowing to provide a response to the requirements of the practice center through the “S+L Booklet”, developed as a methodological proposal for the intermediate practice subject of the Universidad de Las Américas Physical Education Pedagogy career. This instrument takes 15 weeks of weekly development with the proposal, adjustment, development-implementation, monitoring, evaluation and feedback by the students and the teacher of the subject.

## 1. Introduction

Currently, in Latin America, there are high rates of inequality and inequity at the socioeconomic level, which is demonstrated by the disparities in the opportunities to access high-quality education [1]. In this context, education, through educational centers, allows the formation, analysis and reflection of social responsibility [2]. Universities contribute to civic education, social training [3,4] and social justice so that professional training protects teaching strategies that foster their commitment to the community and citizen responsibility [5].

One way in which universities contribute to society is through university social responsibility [6], contributing to the development of the Sustainable Development Goals [7] from the academy. In this sense, Service+Learning (S+L) is an active teaching methodology that allows the university student to connect with society through collaboration networks. It has been based on the constructivist trend, which includes experimental learning, transformative learning, critical theories and historic-cultural theories [8], thus, evidencing the transfer of the learning acquired in their university education for the development of service towards the real need of their community [9]. The S+L, from the Latin American Center for Learning and Solidarity Service (CLAYSS), is defined based on three fundamental features: (a) it is a solitary service destined to attend to the real and latent needs of a particular community; (b) the service is carried out by the students; and (c) its planning allows the integration of the curricular content of learning and research [10]. This way, the S+L establishes an opportunity to develop strategic thinking in a real scenario, being a project that allows the transformation of reality, presenting itself as a challenge for the student body, since it allows the conjugation of personal and community interests based on their requirements [11], highlighting positive perceptions of students for the strengthening of generic skills such as ethics, appreciation of diversity, teamwork, and the moral, values, attitudinal and social dimension [12].

Under this logic, the S+L allows the participation of students in a projective manner, intervening in phases of decision making in relevant situations of the process, and at the same time considering the support and active participation of citizens [13]. This pedagogical model positively evidences the comprehensive training of university students, from teacher training and community service [14], which can even favor labor terms [15,16]. At the same time, it allows the value reinforcement taking into account the high practical educational development with real experiences, which enables the connection of the student body with the community. Throughout the project, this allows the student to reflect on the lived experience of practical experience and collaborative work, where the pedagogical and social dimensions of training interact [17].

Through training focused on community collaboration, it has been shown that it can also provide the skills and experiences that allow professionals to have employment advantages.

## 2. Methods and Context

The Educational Model of the Universidad de Las Americas (UDLA) establishes four dimensions that guide and contextualize the educational work of the Institution. Additionally, within the context of its Institutional Values, the formation of individual and social identity is promoted through Professional Ethics, Community Commitment and Citizen Responsibility [18]. In this way, during the initial teacher training, the students of Pedagogy in Physical Education (in Spanish PEF), through their initial, intermediate, and professional practical training subjects in school contexts [19], can demonstrate these Institutional Values, generating a unification between teaching and the Area of Community Engagement (in Spanish VcM) under the Policy of said unit [20]. In 2020, this methodology was established as an act of connecting teaching actions with the community (Figure 1).

### Development of Teaching Skills through the Service+Learning Methodology

The teacher training process for universities is a great challenge. In addition to having a limited time, it must respond to the various appropriate standards for the country’s development [21,22]. Due to the foregoing, the institutional reflective process can begin with the question: How to decide the importance of one competency compared to others? This answer cannot be standardized since the selection of said competencies will depend on the various demands of society [23], making the decisions of the curricular design of the careers more complex. In this way, the divergence in professional training in education is evidenced in the existing dissimilarity in teaching standards at the international level, where fluctuation is observed in numbers as well as their characteristics [24]. In the case of Chile, during the year 2021, the standards of teacher training have been updated, which captures the substantive elements of the pedagogical action and does not have a prescriptive function on the teaching action, but rather provides criteria for the exercise of teachers in a contextualized way [25]. In the Chilean case, it is the Ministry of Education that, in order to guarantee the quality of higher education, has implemented an evaluation system in such a way that only the universities that achieve said accreditation will be able to impart pedagogy careers [26]. Within this complex system, to guarantee the quality of Initial Teacher Training (in Spanish FID), the Ministry of Education has implemented the so-called “Standards of the Teaching Profession”, which includes pedagogical and disciplinary knowledge for different academic subjects, including Physical Education and Health [25]. These standards demand complex skills such as demonstrating investigative, communication and critical thinking skills.

The international literature has collected experiences in which students in training as practicing teachers declare a low perception of their abilities [27,28], which reinforces the need for higher education institutions to guarantee strategies that allow paying tribute to the demands of the school context.

Based on the aforementioned, the literature has widely confirmed the contribution to training provided by the S+L methodology, among which those associated with social skills and attitudes stand out [29]; additionally, it contributes to professional identity, strengthening metacognitive dimensions [30] and improving the “effective personality” [31]. Likewise, the increased motivation, participation in physical activity, the feeling of belonging and personal achievement such as confidence [32,33,34] also promotes teaching competence by applying motor and expressive games [35]. In addition, Ruiz-Montero et al. [36] have detected a clear contribution to learning, pedagogical value, professional development and the opinion of students in training.

However, despite the benefits declared by the literature, these are not inherently developed, given that records have been found where students show difficulties in acquiring knowledge after applying the S+L methodology [37], which can be explained by the lack of specific training of teachers to implement the methodology [38].

Finally, it is worth mentioning that the methodology receives a variety of denominations, among which ApS, AS, and A-S (in Spanish) stand out, among others, for which reason, the National Service-Learning Network Chile (in Spanish REASE) has determined the use of the abbreviation of S+L (in Spanish A+S) [39], with the latter denomination being the one used in the development of the following proposal.

## 3. Proposal Model

### 3.1. “S+L PEF Booklet” for Methodological Support

In accordance with the guidelines issued by the Curriculum Management Department (in Spanish DGC) and the UDLA PEP Policy, the S+L Methodology is integrated into the University as an active methodology proposal [40], generating integrated dialogical processes between students, teachers, community partners (educational establishments/schools) and the University (UDLA).

For the unification of didactic strategies to guarantee a homogeneous training process between the different sections and guarantee the processes of methodological regularity, the “S+L PEF Booklet” was created as a flexible instrument that allows accompanying both teachers and students in the process of gathering information, and design and evaluation of the projects carried out with the community. Although it has a role of homogenizing the learning experiences of the students, it allows attending to the particularities of each context in which projects are developed, either in students in vocational training or technicians who wish to develop professional skills with respect to the demands of the communities [41].

The “S+L PEF Booklet” was designed considering the proposal of the Manual for solidary teachers and students [10] and the guidelines of Zerbikas [42], which, along with generating strategies to contribute to the community, aim to strengthen the investigative skills of documentary sources, gathering information on territorial needs from both qualitative and quantitative tools, data analysis and interpretation and impact assessment (Figure 2).

Moreover, the proposal aims to contribute to the process of Initial Teacher Training (in Spanish FID) with the development of research skills [43] as well as reflection [44,45], which are emphasized at different milestones throughout the suggested process (Table 1).

### 3.2. Proposal Development

The created document constitutes a concrete pedagogical resource for the application of S+L in a university context that takes for granted the knowledge of the teacher that accompanies the student. In this way, it focuses on accentuating investigative and reflective skills.

For this purpose, the designed plan grants specific sessions to promote the skills of what to do as a teacher [45] without neglecting the linked training purpose of unifying the acquisition of knowledge and its application in real contexts available to society. The booklet is made up of a total of 15 weeks of work, which allow the accompaniment of the processes of intermediate and/or professional pedagogical practices that have a total of 18 sessions per semester (Appendix A). The S+L methodology is used since it allows responding to the real situations of the communities and adjusting to new requirements, proposals or emerging demands from the Community Partner. The proposal is designed so that it can be implemented weekly during autonomous work hours, with synchronous instances for feedback. The weeks are: Week 1 Organization and formation of work teams; Week 2 Community Partner Interview; Week 3 Interview (and transcript) with the Community Partner and Contextual Diagnosis; Week 4 Analysis of interview and institutional documents; Week 5 Presentation of the information to the educational community; Week 6 Intervention design; Weeks 7 and 8 Project implementation; Week 9 Intervention Evaluation; Week 10 Presentation of results and feedback; Week 11 Intervention adjustment; Weeks 12 and 13 Intervention; Week 14 Evaluation, analysis of results and analysis for the projection of the intervention; Week 15 Final presentation of service-learning methodology intervention. The weeks are organized in such a way that the student receives a guarantee of methodological progressiveness (Table 2).

### 3.3. Task Description and Discussion

#### 3.3.1. Week 1–Team Organization and Work Team Formation

This first week of work consists of the organization of the teams for the development of the methodology. Teamwork by affinity was proposed because collaborative work allows social integration, invitation to dialogue and reflection for problem resolution [46,47,48]. In this way, the organization of each team is by affinity because there are already friendship ties and better functioning and efficiency of the workgroup [49,50]. Learning is based on the use of collaborative resources since this contributes to enriching experiences for their performance in the world of work [51]

#### 3.3.2. Week 2–Community Partner Interview

Once the work teams have been formed, the next step is to conduct an interview with the community partner. The community partner is a community, public, private or social organization that is part of the S+L process, which allows the student to connect with their reality through collaboration and communication [52,53,54]. On this occasion, in the case of Physical Education, the community partners are primary (basic) or secondary (middle) educational institutions. Before the students make contact for the development of the interview, the tutor or guide teacher will be the one who establishes the first links or approaches the community partner for the development of the methodology as a collaborative work proposal. Once this management has been carried out by the teacher tutor or guide, all the actors (student, teacher tutor or guide and community partner) meet to develop a proposal that allows them to respond to a real need or requirement of the community partner that may be developed from the work of Physical Education. This action constitutes an instance of learning in which university students approach the territorial demands of the community, thus generating an approximation to the professional and labor context [55]. In addition, this interview must be recorded with prior authorization from the community partner or their representative for the development of Week 3.

#### 3.3.3. Week 3–Community Partner Interview and Transcript and Contextual Assessment

Once the interview corresponding to “Week 2” has been carried out, the students proceed to make the transcript of the interview carried out in the previous work week. In that week, the interview was considered a resource for collecting information because it is a flexible, dynamic process and deepens the dialogue with the interviewee [56,57]. Similarly, the contextual diagnosis allows the reproduction of a real sociocultural context, which allows a better understanding of the student for the diagnosis [58]. Because the observation is relevant within the process, in this way, it is a fundamental part of the lifting of a diagnosis [59].

#### 3.3.4. Week 4–Analysis of the Interview and Institutional Documents

Once the transcription is done, the work team made up of undergraduate students and guided by their tutor professor, perform an analysis of the interview and, at the same time, consult institutional documents (from the educational institution and higher education institution), databases, books or literature corresponding to the topic, for the confirmation of the “Contextual Diagnosis” by the student body and the tutor or guide teacher, to then be presented to the community partner. In this week, the analysis of institutional documents is essential to better understand the analysis of the interview since these documents contain relevant information on the educational project, curriculum, internal regime, among others [60], which allow the student to optimally interpret the educational reality from a critical and reflective point of view, for the acquisition of intellectual autonomy [61].

#### 3.3.5. Week 5–Presentation of the Information to the Educational Community (Community Partner)

Once the interview and the institutional documents have been analyzed for a better interpretation of the educational and social reality, a report of the proposal for the S+L is then developed. The ability to synthesize information must be progressively developed with different methodologies. In this sense, this methodology contributes to this purpose [62]. In the first instance, it can be an approximation for a possible solution and, in this way, respond to the need raised by the community partner. By understanding the relevance and importance of this figure, this proposal may be modified if the educational institution considers it pertinent.

#### 3.3.6. Week 6–Intervention Design

Once the proposal has been presented, and if there are any suggestions, the changes requested by the community partner are taken into consideration, and the intervention is designed. This intervention is based on the steps indicated above and is supported through the curriculum that is part of teacher training to respond to the requirements of the community partner. In this way, the space guaranteed by S+L can even be an opportunity for the professional in training to meet global demands [63].

#### 3.3.7. Weeks 7 and 8–Project Implementation

Under this proposal and understanding the particular characteristics of the requirements of the community partners, the implementation of the intervention is proposed in weeks 7 and 8. However, according to the problems raised by the community partner, the duration of the intervention contained in the proposal may have a variable duration of hours, weeks or months. Furthermore, it should be considered that the service developed through the S+L can generate frustration in the student body, for example, due to very high expectations of the impact of their activity. For this reason, the proposal is a contribution since the interaction with the community is briefly, in this way, it is expected that the student body will generate limited actions with a high sense of effectiveness. Given all the information obtained during the process, it would avoid the development of counterproductive emotions in the training process [64].

#### 3.3.8. Week 9–Intervention Evaluation

At week 9, it is proposed to evaluate and possibly rethink the intervention. For this, a formative evaluation is carried out, which will allow the student to carry out a metacognitive process regarding the proposal. To complete this week’s activities, the student (guided by the teacher) must collect information using a qualitative and/or quantitative technique. This evaluation focuses on the principles of shared formative evaluation, where the student body maintains an active role in this process. In this sense, Fernandez-Garcimartin [65] detected that the perception of teachers and students is positive in this system. 

#### 3.3.9. Week 10–Presentation of Results and Feedback

This week constitutes a feedback space for the student, who must present the results obtained from week 9. Both the professor of the subject and the community partner will be part of it, which will allow objective feedback on the state of progress and impact generated by the intervention. The channels of communication between students in university training with the community are considered relevant in order to avoid a lack of coordination and to allow an interactive process to be maintained with the community [66].

#### 3.3.10. Week 11–Intervention Adjustment

In week 11, by collecting the observations of the processes developed in previous weeks, the student is guided to make an adjustment to their proposal based on the evaluation of week 9, as well as from the feedback obtained in week 10 from the partner, community and teacher of the subject. The adjustments made by the student body favor the feedback of their proposals based on the theoretical-practical elements developed by the subjects, enhancing the reflective component [67].

#### 3.3.11. Week 12 and 13–Intervention

During weeks 12 and 13, students proceed in a timely manner to the development of activities according to the adjustments considered in the educational institution.

#### 3.3.12. Week 14–Evaluation, Analysis of Results, Analysis for the Projection of the Intervention

The activities proposed for this week are the evaluation of the intervention of the direct and indirect beneficiaries by the community partner. Then, the analysis and interpretation of the results are carried out, and finally, the analysis for the projection of the intervention. During this week, the student must propose or create evaluation instruments to collect information about the intervention carried out by the community partner, the direct and indirect participants or beneficiaries of the intervention, and the teacher who guided the process, implemented through the active methodology of the S+L. These instruments must take into consideration gathering information on the degree of contribution of your proposal (community partner), satisfaction (beneficiaries) and process (guide teacher). In this way, the results obtained will allow the assessment of the qualities of the proposal, related to the theoretical and methodological support, and the intervention carried out [68].

#### 3.3.13. Week 15–Final Presentation of Service-Learning Methodology Intervention

Finally, this week the students will make the final presentation of their proposal, in which they must highlight the reflective and projective processes developed in their training and reinforced through the intervention and application of the proposal.

#### 3.3.14. Practical Applications

The curricular proposal developed constitutes an active learning methodology that develops learning based on the demands of citizens, which allows for strengthening the high-quality initial teacher training [69]. This instance allows the future professional to experience the educational reality, a place where it will be developed in the future [70]. In this way, the instrument created allows the development of an S+L project in an undergraduate university context, and as it has been built with the development of generic skills in mind, it can be applied to various subjects.

Regarding its extension, 15 weeks have been considered (an academic semester) for the systematic development of a project that allows students to make available the competencies and skills acquired in their training process, such as reflective thinking, investigation and communication abilities and management for the implementation of actions in the community, to educational institutions.

Literature evidence proposals and guidelines for the development of University Service-Learning projects with a solidarity component allowing children and adolescents to integrate at a social, educational, school and political level, having a broader view such as University Social Responsibility [4]. Although it is a specific instrument, it also allows its adaptation according to the particular characteristics of the community partners, which, in the case of teacher training, corresponds to primary, secondary or special educational institutions.

#### 3.3.15. Future Lines of Research

Considering that the active learning methodology S+L allows the student to collaborate with the educational community, it is suggested in future research, together with the application of the booklet, that evaluative processes be incorporated to allow evidence of the potential impacts that may occur in the educational communities and in university students, whether on a physical, social, educational and/or emotional level. Since it is an S+L experience or project, it can vary and thus, contribute to the development of significant learning.

#### 3.3.16. Strengths and Limitations

The strength of this proposal lies in the creation of a weekly work route evidenced through the booklet. This work route, which is mediated by the teacher and supported by the community partner, allows the strengthening of tools, skills and abilities inherent in the initial teacher training process from the unique perspective of the socio-educational reality that allows the student to interpret during the process. One of the limitations is that an intervention requested by the community partner could have a duration of 60 min, one or several weeks and even one or several months; however, this proposal is structured for the accompaniment of 15 weeks of work with a minimum duration of 60 min. Another limitation of the booklet is that, although it could allow the acquisition and strengthening of skills, competencies and learning tools, these must be deepened in a theoretical and complementary way at the time of the proposal, which could translate into extra chronological hours, that were not initially considered in the teacher training programs. However, if this methodology is implemented in a second opportunity again by the students, times in the work weeks could be reduced to direct theoretical foundation time to other weeks of the proposal.

## 4. Conclusions

The S+L methodology has a great potential which, under a critical and transformative vision of education, allows the student to connect with the real problems in a territory, and in order to attend to them, the students must transfer all their knowledge and skills acquired throughout their training process.

The S+L is an active learning methodology that allows students to transfer the tools, attitudes and knowledge acquired throughout their initial training to real situations in their community; in this way, the community partner raises the problem, and the teacher guides a possible solution to the particular situation presented by the community.

Regarding the usefulness of this booklet, it constitutes a concrete and flexible tool that can guarantee methodological rigor and develop skills typical of teacher training, such as collaborative work, investigative ability and critical thinking, to meet the demands of community partners that under this logic represents the educational establishments.

With the implementation of this pedagogical resource, it is expected to protect the leading role of the student in this process, allowing them to reflect, correct and evaluate their progress individually or in groups, taking into consideration the flexibility of the environment or community where the service is developed.

## Figures and Tables

**Figure 1 ijerph-19-08626-f001:**
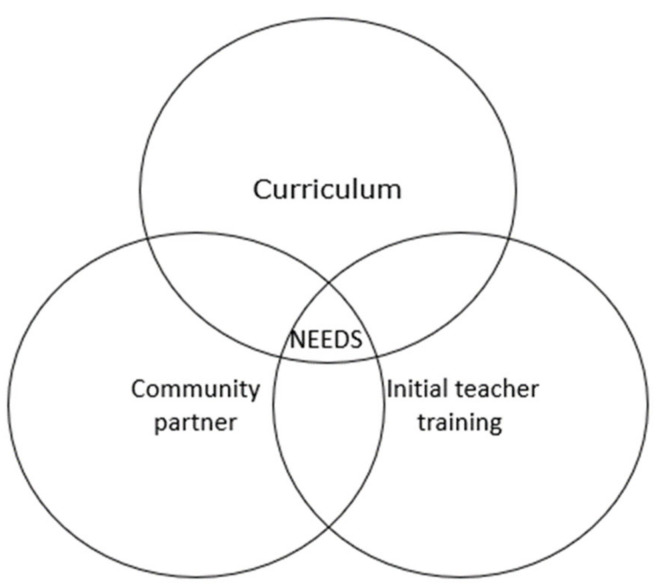
University–Community Partner relationship model. (Authors’ creation).

**Figure 2 ijerph-19-08626-f002:**
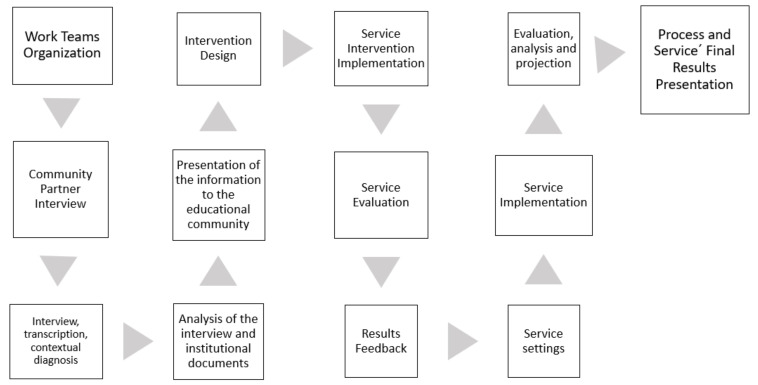
Activities considered for the development of this proposal (Authors’ creation).

**Table 1 ijerph-19-08626-t001:** Development of skills by students.

Contribution to Investigative Skills		Development of Reflection	
Activity	Week(s)	Activity	Week(s)
Design, validation and application of instruments for the collection of information	2-3-6-9-14	Reflection on the community partner	6
Analysis and interpretation of information	4-10-14	Reflection on the own proposal	9
		Reflection on the process (closing)	15

Source: Own elaboration.

**Table 2 ijerph-19-08626-t002:** Scheme of work for the development of the S+L methodology.

Stage	Weeks	Purpose
Organization of the team for collaborative work and identification of the community partner	1,2,3,4,5	Organize and distribute responsibilities among team members and recognize the context and needs of the educational community
Design and implementation of project proposal	6,7,8	Intervention project design validated by the community partner and its application.
Evaluation and presentation of results of the first intervention to the community partner, along with their feedback	9,10	Evaluation process and presentation of results to the community partner for subsequent feedback
First Proposal Adjustment and Reapply Proposal	11,12,13	Adjustment of the intervention, which is generated from the feedback of the community partner
Evaluation and analysis of results	14	Application of final evaluation of the project and subsequent analysis
Project presentation	15	Presentation of project, results, and evidence of the process

(Authors’ creation).

## Data Availability

Not applicable.

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
