# Peer review of "Methodological Proposal for the Accompaniment and Development of Service+Learning Methodology in Universidad de Las Americas in Chile"

_ijerph, 2022, doi:10.3390/ijerph19148626_

Round 1

Reviewer 1 Report

The topic is very interesting, however it is important to developed more in introduction about why to do this investigation, where will be? and why there?

On methods and context I found the information to general. It is no clear who are the public? is it a research apply to all students in the university? the program pedagogy in physical education has the same model of the university?  The study is with the  program pedagogy in physical education ? all students? which students? why this quantity of students?

After in results and creation of the proposal, which proposal?

I didn't find when you said that you will create a proposal, how it was create? that proposal is for all the university or the program pedagogy in physical education? 

I recommend don't speak about teacher training, but pre-service teacher education or in-service teacher education.

Maybe you can re-organize the presentation of the information.  It sounds do me a little confused as it is presented now. 

Author Response

Dear reviewer, we welcome your comments or suggestions. Then we will respond to what you tell us in the attached letter.

Kind regards.

Reviewer 2 Report

The authors propose a methodology, that aims to enable university students to participate in their teaching and learning process. It is a proposal that has not been implemented, so there are no empirical findings.

In methods, the sample should be better explained, for example: (i) are the students of Pedagogy in Physical Education trained to work as Physical Education teachers at primary and secondary schools, or what? (ii) what is the expected sample size and age –range of students/participants?

Underneath Figures 1, 2, and Tables 1, 2, the authors state “Own elaboration”. I suggest to place in parenthesis a phrase like “authors’ creation or authors’ adaptation from xxx”.

In 2.1, the specific social skills and attitudes (related to S+L methodology) should be listed, as well as their usefulness for student training.

I think literature review should be enhanced with more relevant studies.

Regarding the methodology, why do the authors use 2 abbreviations (L+S, S+L) throughout the manuscript?

Appendix A (Line 149) is not included in the paper; clarify is there a section of the supplementary material?

Minor English mistakes, e.g., in Lines 300-301 the sentence is incomplete.

Author Response

(The authors gave the same response as above.)

Reviewer 3 Report

Dear Authors:
Your manuscript needs to be substantially improved.
After reading the manuscript I did not understand why if "Since 2020, the use of this methodology was established as an act of connecting teaching actions with the community" (lines 69-70)", you only present and did not apply your "Methodological proposal".

Some suggestions and comments:

The Title is excessively long.
Summary:
Clarify the objective of the research developed.
In line 18 you write "This instrument (...)" however, it seems to me that it will be "methodological proposal".
The statement: "This methodology allows the student to show all the skills, abilities, tools and theoretical-practical knowledge, since they are placed at the service of the community partner, which in the case of pedagogies translates into pedagogical practice centres. " needs to be revised, "to show all ..." is very pretentious and, at the same time, limiting.

It is not sufficiently clarified/explained:
- why they consider " Learning + Service" instead of "learning service"
- because "Learning + Service (L+S) is an active teaching methodology".
-Section 1. should present the aim of your "Methodological proposal". It also seems to me that sub-section 2.1 should be in section 1.
- line 69 states "Since 2020, the use of this methodology (...)". What is the methodology?

The manuscript is poorly structured:

- section 2 should only refer to the methods, which I consider are not adequately explained. It should adequately describe "Methodological proposal for the accompaniment of an educational and social intervention through the Service+Learning methodology". In fact, it might not even be "Metodos" but "Methodological proposal".

- section 3 is a mess, as there is analysis ... perhaps it does not fit, in this article, a section "3. Results and creation of the proposal ". Maybe just "3. Educational and social intervention through the Service+Learning methodology".
- It should be a separate section "Future lines of research", as well as "Strengths and limitations". It also seems to me that the section "4. Conclusions" does not make sense.

I suggest "final remarks".

Rev

Author Response

(The authors gave the same response as above.)

Reviewer 4 Report

I would like to thank the authors for the interesting article on the methodological proposal for accompanying educational and social intervention through S+L methodology in initial teacher training in physical education pedagogy. I would like to congratulate them for their effort and motivation involved in this research study. The presentation of the research is well documented. A strong point of the article is the bibliographical part including more than 50 footnotes. However, I do have a few comments that need to be answered, as well as a few thoughts that I think would enhance this article:

The abstract should be a single paragraph and should consist of elements such as background, methodology, results, discussion and conclusion. Currently, the abstract is an unstructured description that lacks a clear indication of these elements.

The title of the article is too long. The authors should concentrate on the purpose of the article and keep it as short as possible.

The purpose of the study and the hypotheses should be presented in the introduction in the last paragraph. Currently, the aim is not really highlighted in any part of the article. Therefore, readers will not be fully aware of what the text is about and what the original intentions were. Additionally, in the introduction, the authors should do even more amplification in terms of explaining what S+L is and cite even more studies, especially from English-language publications in reputable scientific journals.

The methodology should be described in sufficient detail to allow others to replicate and use the published results. Currently its assumptions are lost in far more context.

The results of the study are described in detail, but the article lacks discussion. There should be a separate chapter in the article devoted to this, which would compare the results obtained with others and consider the possibility of their implementation in practice. The lack of discussion makes the obtained results unconfirmed and questionable in terms of content.

In general, the article is very interesting, but it lacks numerous elements mandatory for any research manuscript. The authors still have a lot of work ahead of them in the scope of this article, although due to comments primarily of a methodological, technical and structural nature, all of the above tips can be applied. Supplementing the article with the above-mentioned scope will make a real chance for publication in International Journal of Environmental Research and Public Health. I keep my fingers crossed for the final success of the publication.

Author Response

(The authors gave the same response as above.)

Round 2

Reviewer 2 Report

The authors have made the requested revisions

Author Response

Reviewer

Your comments have been considered, we welcome your feedback on the paper.

Kind regards. 

Reviewer 3 Report

Dear authors:

I appreciate your comments. The improvements made to the manuscript have been considerable. However, I am still convinced that section 3 cannot be "Results and creation of the proposal". If we look at your argument, for example at the 1st article https://doi.org/10.3390/ijerph19127366

you will have to agree that your section designation is not correct.
I propose for section 3:
3. Model of the proposal
and subsection 3.3 "General description of the tasks and justification of the purpose of the weeks:"

would be renamed:
3.3 Description of the tasks and discussion

I leave it to you.

Rew

Author Response

(The authors gave the same response as above.)
